# Multi-layer and multi-channel deposition defects and inter-layer control strategies in additive manufacturing materials

Weikang Sun[1], Ming Li[ID][2]*

**1** Case Western Reserve University, Cleveland, United States of America, **2** University of Chinese Academy of Sciences, Beijing, China

* dongxi0724@163.com

## Abstract

Additive manufacturing faces significant challenges in ensuring the quality and dimensional accuracy of components, particularly in complex multi-layer and multi-channel deposition processes. In this study, a novel real-time inter-layer control strategy based on fuzzy logic was proposed. Finite element analysis (FEA) was employed to simulate the temperature field and residual stress distribution in a representative pipeline joint. The simulation revealed that residual stress is significantly higher at the junction of the main and branch pipes compared to the weld zone, with the center of the weld zone exhibiting greater stress than its edges. Deformation analysis indicated a subsidence tendency at the intersection center point (Z-direction: $-3.75 \times 10^{-5}$ mm) and a slight inclination of the branch pipe. A fuzzy logic controller was designed to adaptively regulate the stacking speed of each deposition channel by utilizing the real-time height deviation and its rate of change as inputs. Experimental validation was conducted on three distinct structures: a simple vertical structure, inclined structures (60° and 70°), and a complex funnel structure with varying channel numbers. Results demonstrate that the proposed strategy effectively minimizes deviations: a maximum of 0.13 mm in vertical structures, stable layer growth in inclined structures, and a maximum error of 0.25 mm in the top layer of a funnel structure.

## 1. Introduction

Additive Manufacturing (AM), as a transformative layer-by-layer material deposition approach, has demonstrated unique advantages in fabricating complex components in fields such as aerospace, biomedical, and energy applications [1–3]. Among various AM techniques, Wire Arc Additive Manufacturing (WAAM) is particularly suitable for the rapid prototyping of large-scale metal parts due to its high deposition rate, cost-effectiveness, and favorable mechanical properties [4–6]. However, as the geometric complexity increases, especially in multi-Layer and multi-Channel (MLMC)

**Data availability statement:** All relevant data are within the manuscript.

**Funding:** This research was funded by U.S. National Science Foundation (grant number 2133630) and Chinese Academy of Sciences Open Fund (grant number: 322967812).

**Competing interests:** The authors have declared that no competing interests exist.

deposition, controlling manufacturing accuracy becomes a critical challenge [7,8]. During the sequential layering process, a non-uniform temperature field can induce significant residual stress and thermal distortion, leading to defects such as layer height inaccuracy, localized subsidence, and global inclination [9–11]. These issues severely restrict the broader industrial adoption of WAAM technology.

Current research on precision control in additive manufacturing primarily focuses on two approaches: offline process optimization and online process control [12–15]. Offline methods enhance forming quality by optimizing slicing strategies and process parameters. Nayyori et al. [12] improved the quality of 3D printed components by optimizing slicing algorithms. Oh et al. [13] proposed a direct slicing method to address the high computational complexity associated with complex microporous structures. Combining 2D images with additional features eliminated the complex 3D Boolean operations. The results indicated that the proposed slicing method reduced data size and preparation time, providing significant advantages in implementing microporous structures. Sheng et al. [14] developed an optimization strategy based on extrusion coupled adaptive slicing for simultaneous optimization of the construction direction and layer thickness. Gokhale et al. [15] proposed a deposition framework based on experimental design to achieve an adaptive slicing strategy through inert gas welding. These methods can effectively enhance efficiency during the planning phase, but pre-set parameters are unable to resolve dynamic disturbances caused by heat accumulation and varying heat dissipation conditions during the WAAM manufacturing process. This often leads to unpredictable deviations between the actual formed component and the design model.

In current studies, feedback control strategies for layer geometry in WAAM remain insufficient. Most studies often focus on single-variable control, such as bead width or height [16–18]. For example, Doumanidis et al. [16] developed a multi-variable adaptive controller for the WAAM process, which is based on a generalized one-step-ahead control algorithm. Xiong et al. [17] proposed to use a single neuron self-learning controller to regulate the bead width during WAAM. Besides, Xia et al. [18] developed a Model Predictive Controller (MPC) to control head height during the WAAM process. However, MLMC deposition is a complex, multi-physics, strongly coupled, and parameter-time-varying process, making it difficult to establish an accurate analytical model. The number of deposition channels per layer in MLMC components varies (e.g., pyramid or hourglass structures), and traditional single-variable or fixed-structure controllers fail to adapt such variations.

Fuzzy Logic Control (FLC) has attracted extensive interest during recent decades due to its robustness to model uncertainty and capability of dealing with system constraints [19–22]. This provides a strong theoretical justification for its application in MLMC process control. Nevertheless, existing research employing fuzzy control in welding or additive manufacturing has largely been confined to fixed geometric paths or focused on regulating a single variable, such as bead width. This paper aims to propose and validate an inter-layer control strategy based on fuzzy logic that adapts to variations in the number of deposition channels to minimize deposition defects during the MLMC process.

This paper is organized as follows. First, finite element analysis (FEA) was employed to quantify the temperature field, residual stress distribution, and deformation trends during the MLMC deposition of a pipeline joint. Then, an inter-layer control strategy based on fuzzy logic was proposed. The controller adaptively adjusts the number of deposition channels for each layer in multi-layer structures by using layer height deviation and its rate of change as inputs and deposition rate increment as outputs. Finally, the proposed control strategy was comprehensively evaluated through closed-loop deposition experiments on various geometries, including vertical walls, high-angle inclined structures, and complex hourglass structures. The evaluation focuses on tracking performance and robustness.

## 2. Materials and methods

### 2.1. Simulation of temperature field and residual stress in MLMC deposition

The substantial heat input inherent to MLMC deposition is a primary cause of geometric defects, including sinking and tilting. To predict these defects, a coupled thermal-mechanical finite element analysis was conducted to simulate the transient temperature field and subsequent residual stress evolution. The welding heat source exhibits non-uniform, transient, and moving characteristics, which must be accurately represented in the numerical model. The total heat generation during WAAM comprises multiple components: arc heat, latent heat of phase change, deformation heat, and resistive heating [23]. For modeling practicality, the latent heat of phase change and resistive heating were incorporated into the effective arc heat input, given the difficulty in their precise quantification. The deformation heat was neglected due to its comparatively minor contribution. The dominant arc heat [24] is typically represented in simulations by a welding heat source model. Common numerical representations include the Gaussian surface heat source (GHS), the double ellipsoidal volume heat source (DEHS), and the volumetric heat source. The models of GHS and DEHS are illustrated in Fig 1.

The GHS model assumes that the distribution of heat flux density within the arc coverage area on the component surface can be described using a normal distribution function, applying heat input only to the surface [25]. The DEHS model uses a semi-oval distributed volume heat source to describe the moving heat source model during deep penetration surface welding of butt welds. Within a single semi-oval volume heat source, the heat flux density follows a normal distribution [26]. Alternatively, a volumetric heat source directly applies heat input as a body load to the weld element ensemble, which is widely used in numerical simulations of irregular welds, as expressed in Equation (1):

$$HGEN = \frac{\eta \cdot U \cdot I}{A \cdot T_w \cdot Dt}$$

(1)

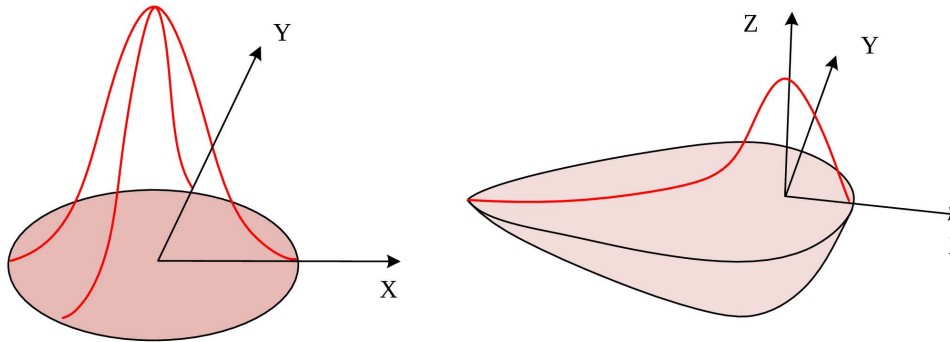

(a)Gaussian heat source model (b)Double ellipsoid heat source model

**Fig 1. Schematic diagram of GHS and DEHS.**

where $\eta$ denotes power efficiency, $U$ the welding voltage, $I$ the welding current, $A$ the cross-sectional area of the weld seam, $T_w$ the welding time, and $Dt$ the step-by-step time of welding load. For irregular weld geometry, accurately defining the heating surfaces or ellipsoidal volumes for GHS or DEHS models is challenging. Therefore, a volumetric heat source model was adopted; the heat input is directly applied as a body load to the weld element ensemble. In addition to this heat input, various heat loss mechanisms need to be considered, including heat conduction, enthalpy transfer, radiative heat transfer, and convective heat transfer. Among these, heat conduction transfers thermal energy from high temperature to low temperature following Fourier's law, as shown in Equation (2):

$$q^* = -\lambda \frac{\partial T}{\partial n}$$

(2)

where $q^*$ is the heat flux density of the heated object (J/(mm2·s)). $\lambda$ is the thermal conductivity (J/(mm·s·K)). $\frac{\partial T}{\partial n}$ is the temperature gradient ($K$/mm). Radiation heat transfer is described by Equation (3):

$$q_r^* = \varepsilon C_0 T_s{}^4$$

(3)

where $q_r^*$ is the heat flux density (J/(mm2·s)), $\varepsilon$ is the emissivity coefficient, $T_s$ is the surface temperature of the solid, and $C_0$ is the constant ($5.67 \times 10^{-14}$ J/(mm2·s·K4)) when $\varepsilon = 1$. Convective heat transfer refers to the heat transfer process that occurs between a fluid and a solid wall when they are in direct contact at different temperatures [23]. Convective heat transfer following Newton's law is given in Equation (4):

$$q_c^* = -\alpha_c(T - T_0)$$

(4)

where $q_c^*$ represents the heat flux density (J/(mm$^2$·s)) of the solid surface in contact with gas or liquid. $\alpha_c$ is the convective surface heat transfer coefficient (mm$^2$·s·K), which relies on the density and viscosity of the surrounding medium, the characteristics and shape dimensions of the solid material, as well as temperature differences. $T_0$ is the fluid temperature ($K$). A pipeline joint structure was selected as a representative case for simulating the temperature field and residual stress in MLMC deposition processes, given its prevalence in practical welding applications. Both the main and branch pipes in this model are Q345 carbon steel. The pipeline joint structure is illustrated in Fig 2.

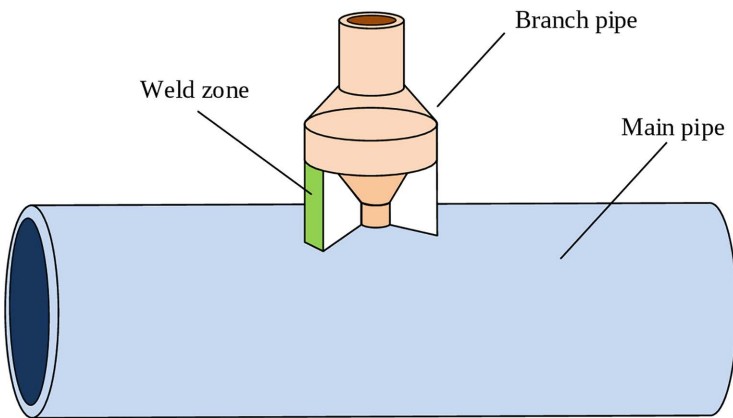

**Fig 2. Schematic diagram of pipeline joint structure.**

The finite element simulations were performed using ANSYS software, with parametric modeling implemented through the ANSYS Parametric Design Language (APDL). First, based on the actual dimensions of the workpiece, the three-dimensional model incorporating both the base structure and the weld seam geometry was established in the APDL environment. To achieve an optimal balance between computational accuracy and efficiency, a hybrid meshing strategy was adopted. The branch pipe was treated as a discrete entity, with the volume representing the direct interfacial contact between the main and branch pipes designated as a transition zone. Taking into account that the steepest temperature gradients occur within and near the weld seam, this critical region was discretized using a refined hexahedral mesh. The surrounding transition zone was filled with a tetrahedral mesh to facilitate connection. In contrast, regions located far from the heat-affected zone, where thermal variations are minimal, were assigned a relatively coarse hexahedral mesh.

After completing the model establishment and mesh division, the "birth and death" element technique in ANSYS was used to deactivate all elements in the weld area initially. Then, based on APDL cyclic commands, the welding heat source was moved through the overload step function, gradually activating the weld area elements and applying loads. Before the calculation, the necessary boundary and initial conditions were defined: a convective heat transfer coefficient was applied to all exposed surfaces with an ambient temperature of 298.20 K; fixed constraints were imposed at both ends of the 600-mm-long main pipe to represent rigid supports; and a uniform initial temperature field of 150°C was set to simulate the preheating state before deposition. Additionally, a cooling stage is implemented following the deposition stage to allow stress redistribution. The simulation modeling and mesh division results of pipeline joints are shown in Fig 3.

## 2.2. Inter-layer control strategy based on fuzzy logic

As indicated in the previous section, in the actual WAAM process, the deposition path exhibits deviations in the height direction due to non-uniform temperature field distributions. When accumulated over multiple layers, these deviations can critically affect the overall dimensional accuracy of the fabricated component. Therefore, implementing effective inter-layer height control is crucial. The forming process of MLMC structure deposition channel is highly complex, and it is difficult to derive an accurate mathematical model to quantitatively describe its forming behavior. Fuzzy logic control (FLC) provides a viable alternative, as its construction based on fuzzy concepts and logic without requiring an exact analytical model of the controlled process. Therefore, an inter-layer control strategy based on fuzzy logic inference was designed.

Fuzzy control employs a fuzzy inference machine as its core to solve fuzzy and uncertain problems in complex systems based on fuzzy rules, thereby obtaining fuzzy control variables [19]. The key to the inference process of fuzzy

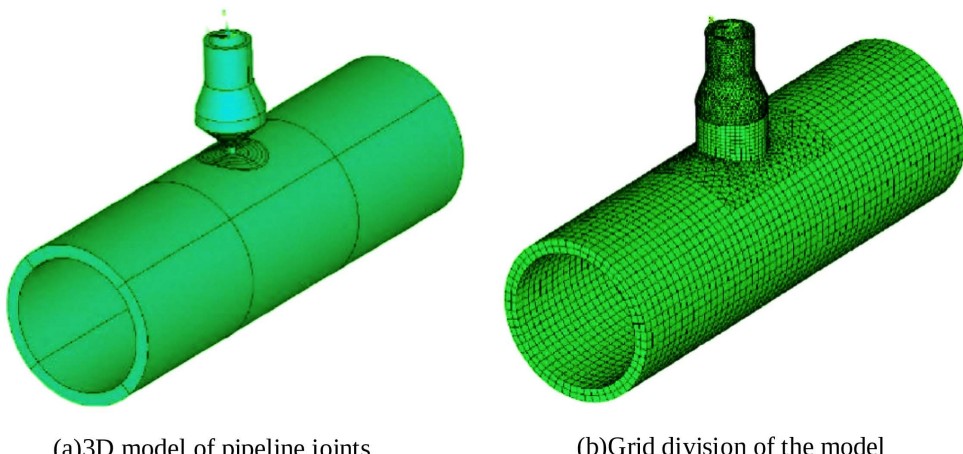

(a)3D model of pipeline joints          (b)Grid division of the model

**Fig 3. Simulation modeling and grid partitioning results.**

controllers lies in the fuzzy rule table and control variables. To design a fuzzy logic controller suitable for MLMC deposition forming, a fuzzy controller was programmed in Matlab environment. The judgment criteria for the height deviation at each position are divided into seven levels: extremely high, very high, high, normal, low, very low, and extremely low. The corresponding fuzzy subsets are PB, PM, PS, Z, NS, NM, and NB. The fuzzy rule table selected for the study is presented in Table 1.

After comprehensive consideration, the actual range of the height deviation of the deposition channel was set to −3 mm~3 mm, the change in the actual range of the height deviation is set to [−1,1], and the control range for the speed increment was set to [−2, 2]. The core principle of the proposed inter-layer control strategy is to adjust the stacking velocity for each deposition channel. Specifically, before stacking the deposition channels, a set of optimal velocity differences $\Delta v$ is calculated based on the number of deposition channels and added to the initial stacking velocities set for each deposition channel during the planning phase.

Assuming the deposited layer inside a certain part is composed of $n$ deposited channels, the preset forming height of the $n$-th welding channel in the $m$-layer is $H_s(m, n)$ after stacking, and the actual forming height is $H_r(m, n)$. The actual formed height deviation $e(m, n)$ and the change in height deviation $\Delta e(m, n)$ at that position are given by Equation (5):

$$\begin{cases} e(m, n) = H_r(m, n) - H_s(m, n) \\ \Delta e(m, n) = e(m, n) - e(m-1, n) \end{cases}$$

(5)

Directly applying control based on $\Delta v$ may cause instability due to the inconsistent stacking speed of different layers in the deposited layer. Therefore, this study uses the change amount $\Delta\Delta v$ of velocity increment for control, which is accumulated layer by layer to determine the optimal velocity offset. By inputting $e(m, n)$ and $\Delta e(m, n)$ into the fuzzy logic inference machine, the change amount $\Delta\Delta v(m+1, n)$ of the accumulation velocity increment of the next layer at that position of the weld deposit is obtained. This incremental change is added to the velocity increment $\Delta v(m, n)$ calculated for the current deposition channel to yield the velocity increment $\Delta v(m+1, n)$ for the next layer. Finally, the stacking speed of the deposition channel in the next layer is modified, as expressed in Equation (6):

$$v_r(m+1, n) = v_s(m+1, n) + \Delta v(m+1, n)$$

(6)

where $v_r(m+1, n)$ and $v_s(m+1, n)$ denote the actual stacking speed and preset stacking speed of the $n$-th deposition channel in the $m$-th, respectively. For a deposited layer composed of $n$ deposited channels, $n$ sub-controllers in parallel to compute the control parameters for each channel, thereby regulating the forming height in all layers. The calculated speed increment $\Delta v(m, n)$ is directly applied during the stacking process. The change in height deviation is derived from the forming heights of two consecutive deposited layers. Before depositing the first two layers, the initial speed increment for each sub-controller is set to zero

**Table 1. Fuzzy rule table.**

| Speed increment | | Height deviation | | | | | | |
|---|---|---|---|---|---|---|---|---|
| | | NB | NM | NS | Z | PS | PM | PB |
| The change in height deviation | NB | NB | NB | NM | NM | NS | NS | Z |
| | NM | NB | NM | NS | NS | NS | Z | PS |
| | NS | NM | NS | NS | NS | Z | PS | PS |
| | Z | NM | NS | NS | Z | PS | PS | PM |
| | PS | NS | NS | Z | PS | PS | PS | PM |
| | PM | NS | Z | PS | PS | PS | PM | PB |
| | PB | Z | PS | PS | PM | PM | PB | PB |

(i.e., $\Delta v(1,1) = \Delta v(1,2) = ... = \Delta v(1,n) = 0$ and $\Delta v(2,1) = \Delta v(2,2) = ... = \Delta v(2,n) = 0$). The first two layers are directly deposited using pre-designed stacking parameters for each deposition channel. Then, detection and deviation calculation were carried out based on the forming height at each position.

The above control strategy, characterized by relatively independent sub-controllers, is well-suited for simple vertical MLMC structures. However, practical applications often involve more complex geometries, such as pyramid structures where the component width varies with height, leading to a variable number of deposition paths in different layers. To address this challenge, an extended control strategy capable of handling dynamic changes in the number of controlled variables was proposed. According to the fundamental control rules, each sub-controller determines the adjustment to the preset stacking speed for the next layer based on three key parameters: the height deviation $e$ of the previously formed layer, the speed adjustment $\Delta v$ applied to the same channel in the previous layer, and the change deviation $\Delta e$ of the forming deviation in the height direction. Variations in the number of channels in different layers along the height direction result in two situations: increase and decrease. Therefore, these two cases are analyzed separately, as shown in Fig 4.

For the situation where the number of channels decreases with increasing height (i.e., pyramid structure), it is necessary to merge the control parameters in the lower-layer sub-controllers. The procedure involves two primary steps: first, the average height deviation and average deviation change are computed from the two underlying deposition channels, and these averaged values are processed through the fuzzy logic inference system to determine the new incremental speed adjustment $\Delta\Delta v(m+1,n)$. Second, the speed adjustment for the merged channel is calculated by combining this new incremental adjustment with the inherited speed adjustment from the lower-level sub-controllers. The control amount $\Delta v(m+1,n)$ of the deposition channel speed, derived from this merging process, is given by Equation (7):

$$\Delta v(m+1,n) = \frac{\Delta v(m,n) + \Delta v(m,n+1)}{2} + \Delta\Delta v(m+1,n)$$

(7)

Finally, the actual stacking speed is obtained by summing $\Delta v(m+1,n)$ and $\Delta v_s(m+1,n)$. For the case where the number of channels increases with height (i.e., inverted pyramid structure), the control strategy of the middle deposition channel also needs to account for the three parameters from the adjacent two deposition channels. The height of the upper edge deposition channel is influenced solely by the forming height of the edge channels in the overlapped layer. Therefore, the parameters in the sub-controller are consistent with the controller of the lower edge deposition channel. The calculation of the control variable follows the same method established for the simple vertical MLMC structure. In summary, for cases where

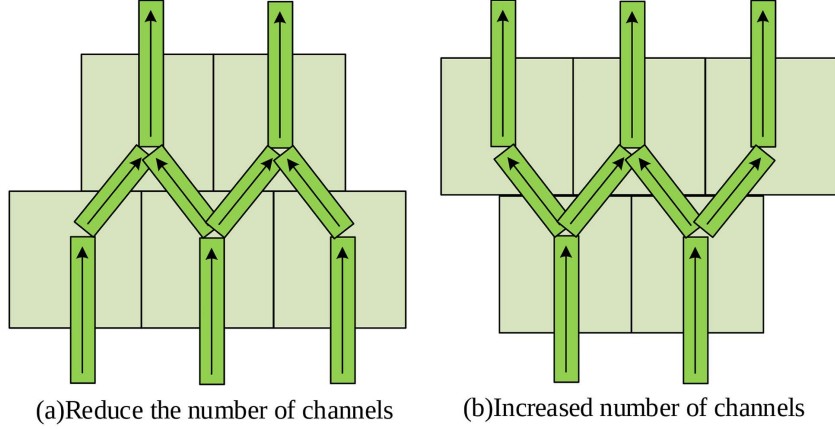

(a)Reduce the number of channels (b)Increased number of channels

**Fig 4. Schematic diagram of reducing and increasing the number of channels.**

the number of channels decreases along the height direction, the number of sub-controllers should be reduced accordingly. When the number of channels increases, the number of sub-controllers should be increased accordingly. This adaptive framework ensures precise regulation of the stacking speed for each deposition channel in different deposition layers.

## 3. Results

### 3.1. Analysis of temperature field and residual stress simulation results

To characterize the thermal evolution during MLMC deposition, a layered deposition approach was implemented along the radial direction of the branch pipe. Within the established simulation model, monitoring points were designated at the shoulder (Point A) and the abdomen (Point B) of the main pipe, as illustrated in Fig 5.

The temperature-time curve changes of points A and B are shown in Fig 6. It can be seen that the temperature at both points initially increased and then decreased over time. Point A reached a maximum temperature of approximately 331 K,

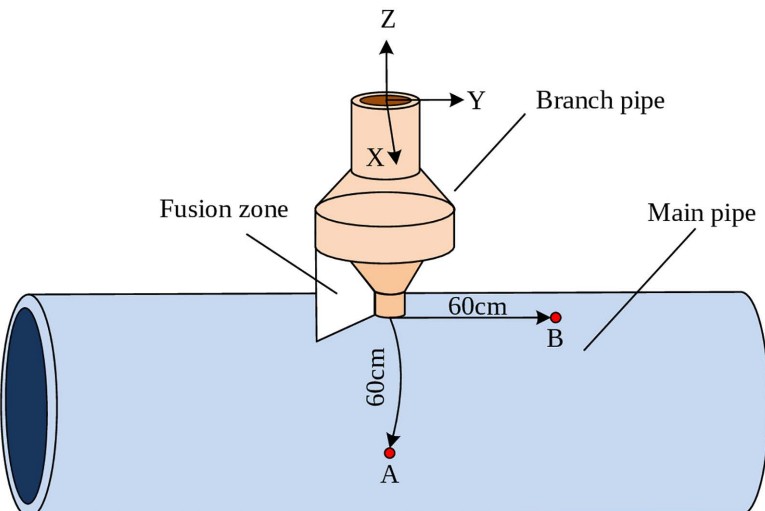

**Fig 5. Layered and point sampling diagram.**

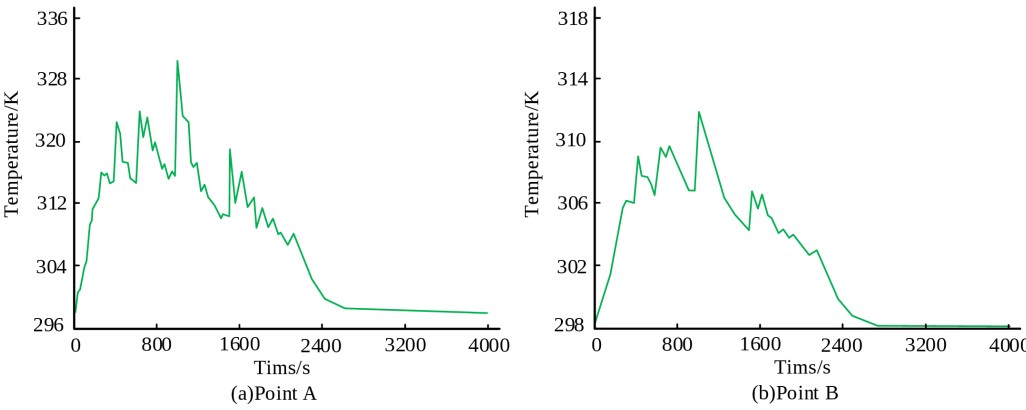

**Fig 6. Temperature time curves of points A and B.**

which was about 18 K higher than the peak temperature recorded at Point B. This temperature differential indicates that the heat input transferred to the shoulder of the main pipe is higher than that transferred to its abdomen.

To further assess the thermal trend at the junction, four equidistantly distributed points C, D, E, and F were selected on the intersecting surface between the main and branch pipes. The temperature-time curve changes of these four points are shown in Fig 7. All four points exhibited significant thermal fluctuations during the initial 2400 seconds of the process and stabilized thereafter. The temperature changes at points C and D were closely aligned, with peak temperatures ranging between 700 K and 750 K (Fig.7a,b). Similarly, points E and F exhibit comparable behavior, with higher peak temperatures between 750K-880K (Fig.7c,d).

The residual stress distribution was analyzed by selecting specific points within the weld zone. Four points, c1, d1, e1, and f1, were located at the center of the weld zone corresponding to C, D, E, and F. Complementary points c2, d2, e2, and f2 were positioned at the edges of the weld zone. The residual stress values are displayed in Table 2. A clear stress gradient was observed; the residual stress at the junction of the main and branch pipes (Points C–F) was markedly higher

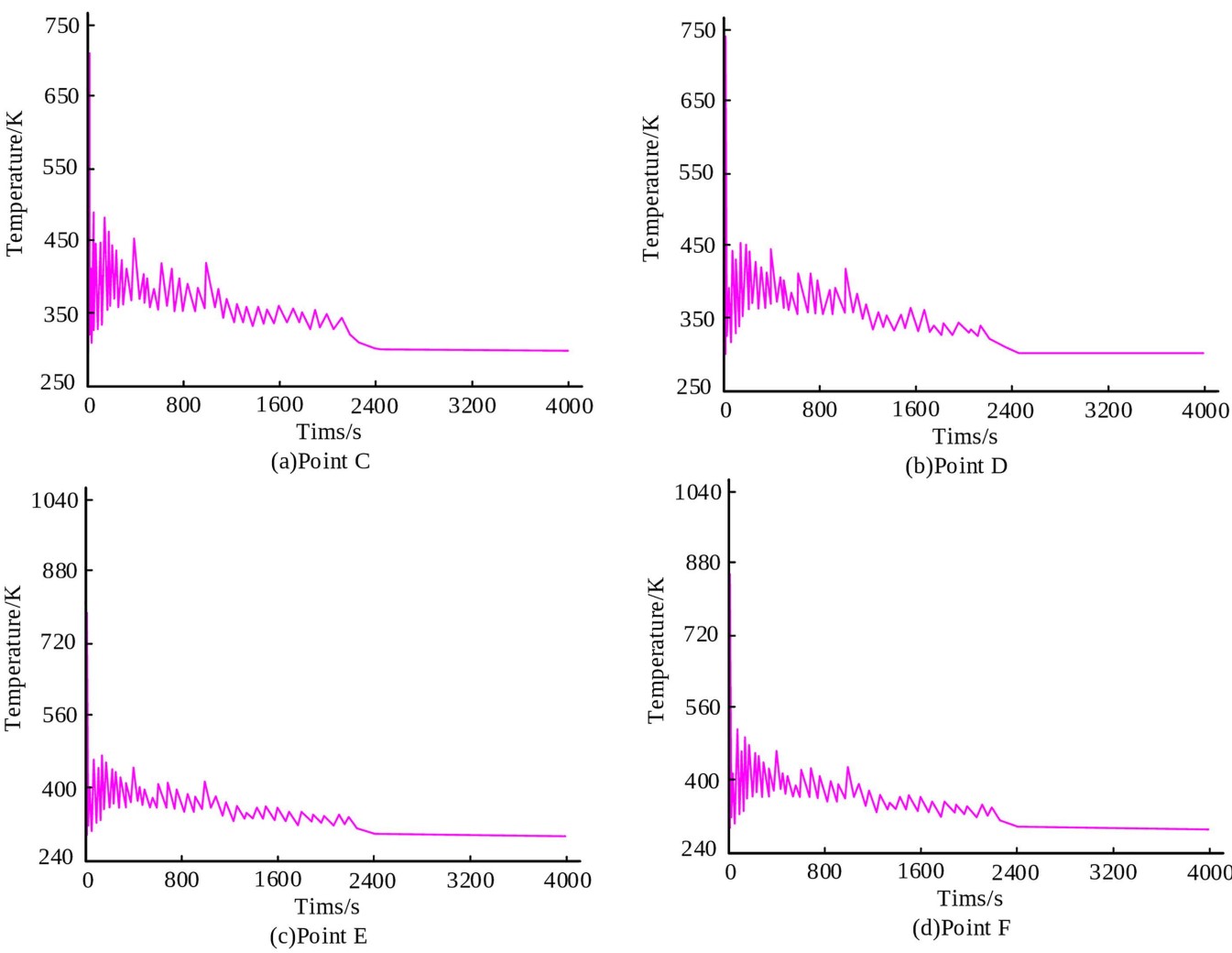

**Fig 7. Temperature time curve of four points.**

**Table 2. Residual stress values at each point.**

| Point | Residual stress/MPa | Point | Residual stress/MPa | Point | Residual stress/MPa |
|-------|---------------------|-------|---------------------|-------|---------------------|
| C | 146 | c1 | 88 | c2 | 49 |
| D | 158 | d1 | 67 | d2 | 39 |
| E | 168 | e1 | 56 | e2 | 41 |
| F | 130 | f1 | 34 | f2 | 45 |

than that within the weld zone (Points c1–f1, c2–f2). The maximum stress of 168 MPa was recorded at Point E, while the minimum of 130 MPa occurred at Point F. Furthermore, the residual stress at the center of the weld zone exceeded that at the edge of the weld zone (e.g., 49 MPa at c2). It demonstrated that the most significant residual stress is generated at the root of the weld seam, with comparatively lower stress levels at the edge of the weld seam.

To evaluate the subsidence and inclination of branch pipes during the welding process, the vertical (Z-direction) displacement of the intersection center (Point H) was monitored, along with the lateral (X and Y-direction) displacements of a point at the top of the branch pipe (Point I). The displacement of the branch pipe is shown in Fig 8. During the initial deposition stage, the displacement of point H in the Z direction rises significantly, reaching a maximum of $8.75 \times 10^{-5}$ mm (Fig.8a). As the deposition layer increased and the heat source moved away, the displacement of point H in the Z direction first decreased and then increased in the opposite direction. The final displacement stabilized at $-3.75 \times 10^{-5}$ mm, indicating that the branch pipe underwent subsidence. From Fig 8(b), during the deposition process, the point I in the Y continuously fluctuated and eventually stabilized at $-0.4 \times 10^{-5}$ mm, indicating that the branch pipe had a certain degree of inclination. In the X-direction, during the initial stage of deposition, the point I also increased significantly, reaching a maximum of $4.01 \times 10^{-5}$ mm. As the deposition layer increased and the heat source moved away, the point I first decreased and then increased in the opposite direction, finally stabilizing at $-0.12 \times 10^{-5}$ mm, indicating that the branch pipe was inclined (Fig. 8c). These results demonstrate the tendency of subsidence and inclination in branch pipe structures under MLMC deposition conditions.

### 3.2. The effectiveness of the inter-layer control strategy

To validate the effectiveness of the proposed inter-layer control strategy based on fuzzy logic inference, an experiment was conducted on a vertical MLMC structure comprising 20 layers, each layer containing 4 deposition channels. Under constant processing parameters of 150A current, 22V voltage, and 6 mm/s speed, the deposition was carried out with a width of 7.33mm and a height of 2.3mm. The distance between adjacent deposition centers is 5.41mm, the offset distance of overlapping deposition paths is 0.23mm, and the layer height is 2.08mm.

A reference deposition was first conducted using conventional open-loop methods. Under this parameter, the first deposition path becomes gradually shorter with the increase of layers, resulting in areas of material loss. Following the implementation of the inter-layer control strategy, the stacking speed and height deviation of each deposition channel are shown in Fig 9. Analysis of Figs 9(a)-(d) reveals that the overall forming deviation of the component fluctuates stably around zero, with a maximum deviation of 0.13mm. The results demonstrate that even under suboptimal initial parameter conditions, the inter-layer control strategy based on fuzzy logic inference effectively maintains the forming height close to the target value, validating the feasibility of the method.

To investigate the effectiveness of the inter-layer control strategy based on fuzzy logic inference for inclined structure deposition, an MLMC deposition experiment was conducted on a 60° inclined structure with constant width, using an inter-layer offset of 1.23mm. The processing parameters, the layers of the component, and the deposition channels for each layer remain unchanged. The stacking speed and height deviation of each deposition channel are shown in Fig 10. As illustrated in Figs 10(a)-(d), under the proposed inter-layer control strategy, the height of each position of the

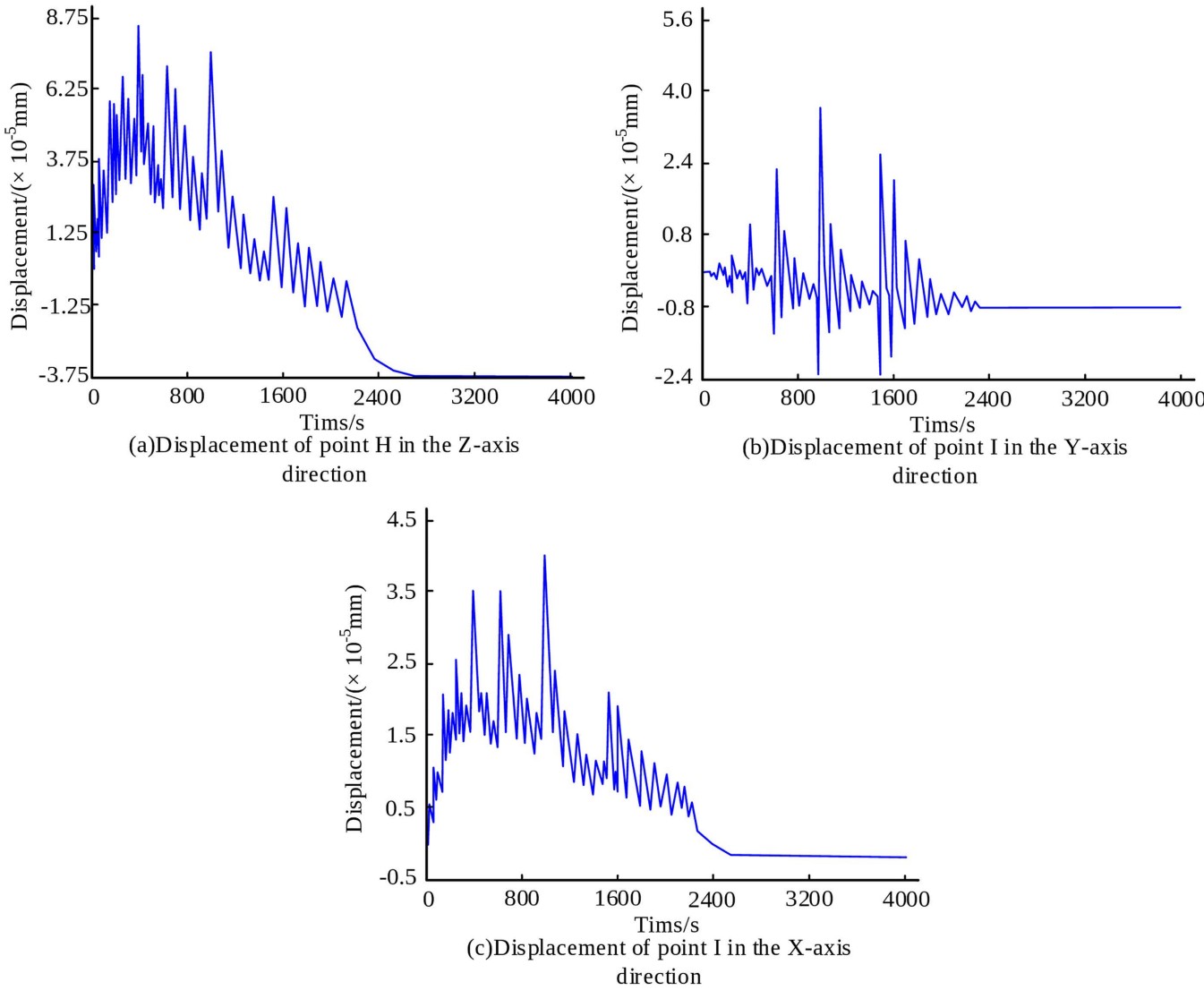

**Fig 8. Displacement monitoring of branch pipes.**

60° inclined and constant width structure steadily increased. The overall forming deviation of the component fluctuates around zero.

The control strategy was also applied to a 70° inclined structure with constant width. The stacking speed and height deviation of each deposition channel are shown in Fig 11. From Figs 11(a), (b), (c), and (d), it is evident that the overall forming of the part reached the preset height, achieving the expected control purpose. This confirms that the proposed inter-layer control strategy performs effectively in the MLMC deposition of inclined and constant width structures.

To evaluate the performance of the proposed method in controlling structure deposition forming with width variations in height, experiments were conducted on the MLMC deposition control of funnel structures. The component consists of 17 deposited layers. The number of deposition channels per layer varied as follows: layers 1–2 contained 6 channels; layers 3–4 contained 5 channels; layers 5–7 contained 4 channels; layers 8–10 contained 3 channels; layers 11–13 contained 4

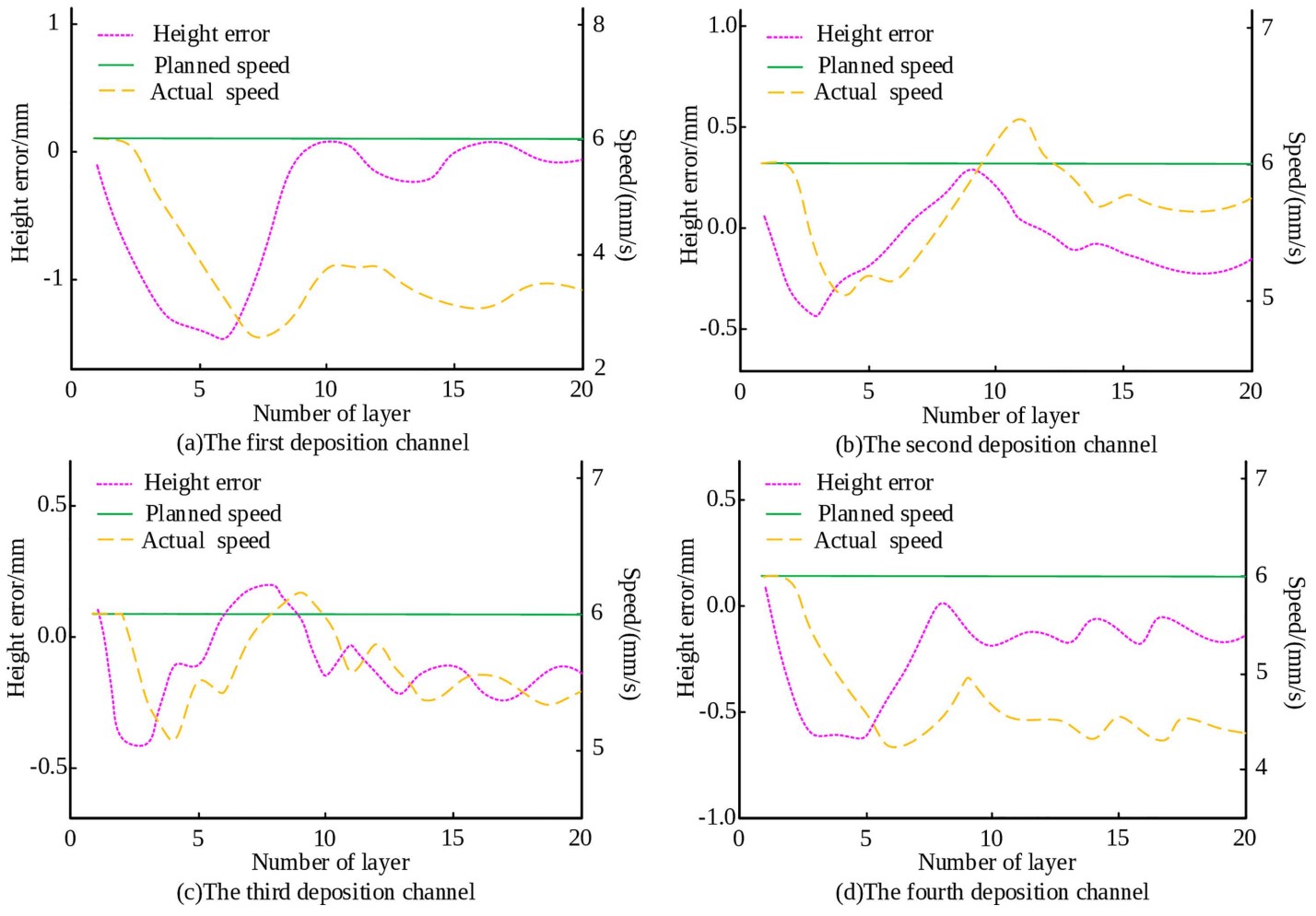

**Fig 9. The stacking speed and height deviation of each deposition channel.**

channels; layers 14–15 contained 5 channels; and layers 16–17 contained 6 channels. The absolute forming error of each deposition channel is shown in Fig 12. It can be observed that the maximum error of each deposition channel in the 17th layer was 0.25mm. Notably, the edge deposition channels exhibit no significant subsidence and inclination, confirming the achievement of the preset control effect.

The actual stacking speed of each deposition channel is shown in Fig 13. As shown, the minimum stacking speed utilized during deposition was 2.51mm/s. Combined with Fig 12, at this speed, no significant subsidence or inclination is observed in the edge deposition channels. The overall formation of the component corresponds to the established design modes. The results indicate that the proposed method maintains effective control performance in structure deposition forming with width varying in height.

## 4. Discussion

To effectively regulate the deposition height in MLMC arc additive manufacturing processes, this study employed numerical simulation to reveal the temperature field and residual stress distribution patterns during deposition. Based on this foundation, a fuzzy logic-based inter-layer control strategy was proposed and validated through systematic experiments.

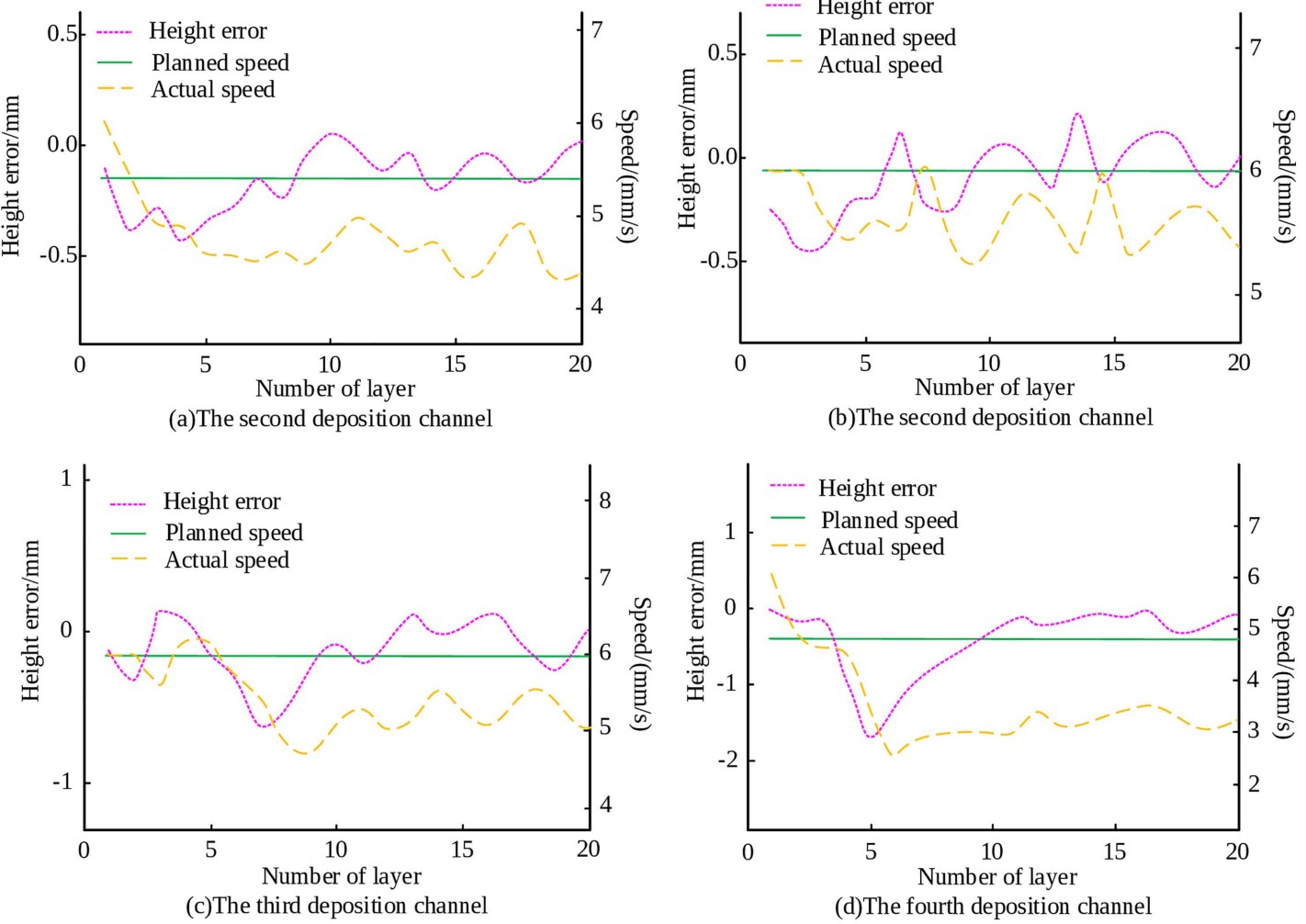

**Fig 10. Accumulation velocity and height deviation of each deposition channel in a 60° inclined structure.**

The simulation results reveal distinct thermal and mechanical behaviors during the deposition process. From the results, both the shoulder point A and abdominal point B of the main pipe exhibit a temperature trend that first increases and then decreases. The peak temperature at Point A reaches 331 K, approximately 18 K higher than Point B. The surface residual stresses at the junction between the main pipe and branch pipe are significantly higher than those in the weld zone, ranging from 130 MPa to 168 MPa. Furthermore, the residual stress at the center of the weld zone is markedly higher than that at the edge of the weld zone. Deformation analysis indicated a subsidence tendency at the intersection center point, with a maximum Z-direction displacement of $8.75 \times 10^{-5}$ mm, and the final displacement remained stable at $-3.75 \times 10$-5mm. The displacements of the branch pipe top in the Y and X directions stabilized at $-0.4 \times 10^{-5}$ mm and $-0.12 \times 10^{-5}$ mm, respectively, indicating an extent of inclination in the branch pipe.

The experiment validation confirmed the effectiveness of the proposed control strategy under various structural configurations. For the simple vertical MLMC structure, the overall forming deviation is constrained within ±0.13 mm, oscillating consistently around the target value. The height of the 60° and 70° inclined and constant width structures, the controller ensured stable height growth at all positions, with overall deviations fluctuating near zero. Most notably, the funnel

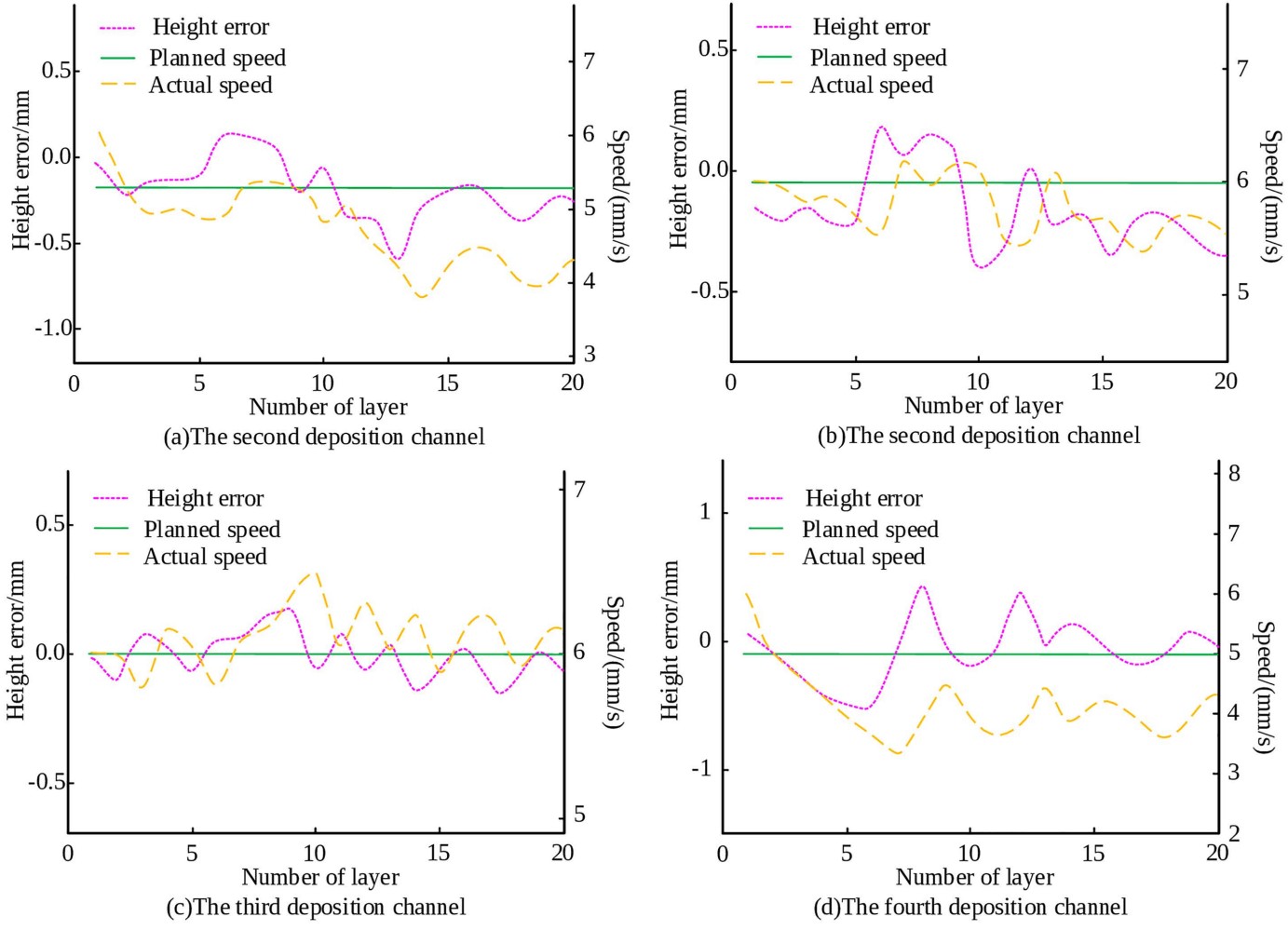

**Fig 11. Accumulation velocity and height deviation of each deposition channel in a 70° inclined structure.**

structure with variable cross-section exhibited a maximum error of only 0.25 mm in the top layer, with no significant edge subsidence or inclination, confirming excellent flatness control.

Unlike material modification approaches that enhance intrinsic properties through alloying elements [12], the proposed strategy actively compensates for deformation caused by uneven temperature field and residual stress by adaptively regulating the stacking speed. Furthermore, previous research focused on computational efficiency through slicing algorithm optimization [13–15], this study emphasizes real-time adaptive control during the deposition process. In practical applications, the method provides a valuable technical support for controlling the precision of complex components in rapid manufacturing and remanufacturing. It can be directly integrated into existing intelligent additive manufacturing systems to improve process stability and product consistency.

While the current study demonstrates promising results, some limitations should be acknowledged. The control strategy was validated under specific deposition conditions and material (Q345 steel). Its performance with other materials (e.g., aluminum alloys or high-strength steels) or under significantly different thermal regimes deserves further investigation. Additionally, the controller's parameters were tuned for the experimental setup described; generalizing these parameters for broader industrial applications may require additional adaptive mechanisms.

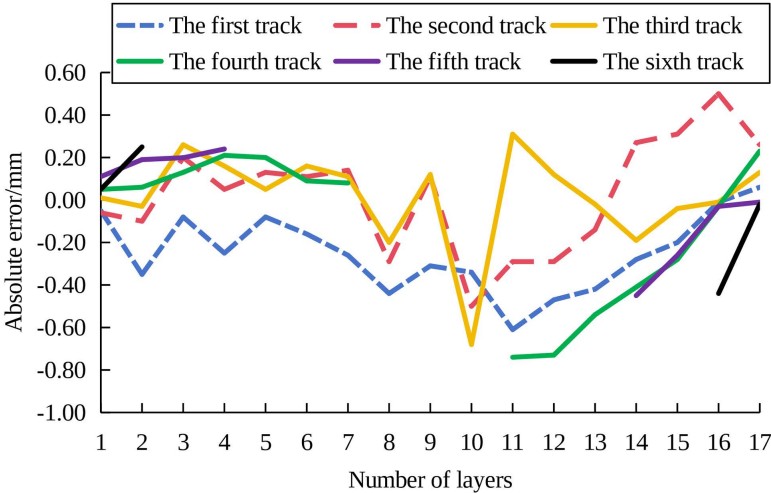

**Fig 12. Absolute error of each deposition channel.**

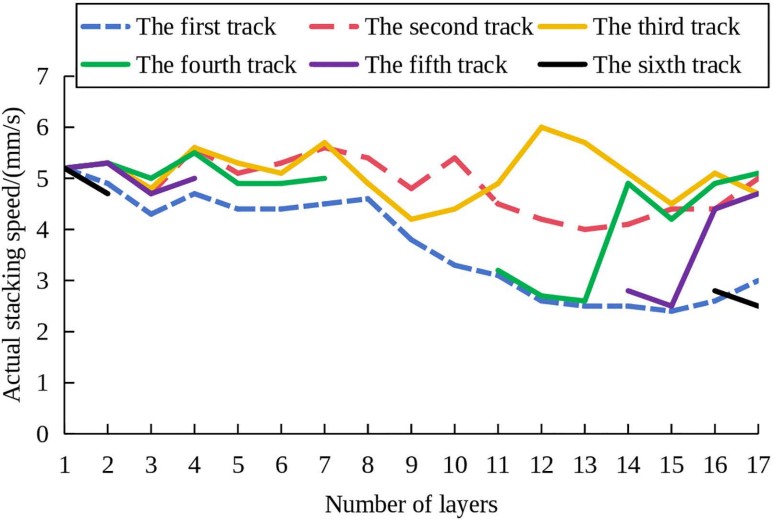

**Fig 13. Actual stacking speed of each deposition channel.**

## 5. Conclusions

This study addressed the precision control challenge in multi-layer and multi-channel additive manufacturing by proposing a feasible and efficient control strategy. The developed fuzzy logic-based inter-layer control system demonstrates effective regulation across successive deposition layers and maintains notable adaptability and robustness under different geometric structures.

Finite element simulation of a representative pipeline joint revealed crucial thermal behavior during MLMC deposition. The results quantified significant residual stress concentrations at structural junctions (up to 168 MPa) and identified subsidence ($-3.75 \times 10^{-5}$ mm) and inclination tendencies in branch pipes. To adaptively compensate for the identified thermal deformation, a fuzzy logic controller (FLC) was designed to dynamically adjust the deposition speed of each channel

based on real-time layer height deviation and its rate of change. Meanwhile, the controller is able to adapt structures with variable channel counts (pyramidal and funnel geometries) without requiring precise analytical models of highly nonlinear processes. Experimental validation confirmed the effectiveness of the proposed controller under various structural configurations. For a simple 20-layer vertical structure, the maximum height deviation was constrained to 0.13 mm. For 60° and 70° inclined structures, the controller ensured stable layer growth, with overall deviations fluctuating near zero. Furthermore, for a complex funnel structure with dynamically varying channel numbers, the strategy maintained the top-layer error within 0.25 mm without significant edge subsidence or inclination.

It should be noted that this study primarily focuses on controlling geometric defects during the MLMC deposition process. Subsequent precision machining operations may introduce additional deformation. Future research should therefore extend this control framework to the complete manufacturing process, integrating adaptive compensation strategies for post-deposition machining processes to further enhance the overall quality of additively manufactured components.

## Author contributions

**Conceptualization:** Ming Li.

**Funding acquisition:** Ming Li.

**Software:** Weikang Sun.

**Writing – original draft:** Weikang Sun.

**Writing – review & editing:** Ming Li.

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
