## [Decision Letter · Decision Letter 0]

17 Dec 2025

Dear Dr. Li,

Thank you for submitting your manuscript to PLOS ONE. After careful consideration, we feel that it has merit but does not fully meet PLOS ONE’s publication criteria as it currently stands. Therefore, we invite you to submit a revised version of the manuscript that addresses the points raised during the review process.

We look forward to receiving your revised manuscript.

Kind regards,

Mithilesh K. Dikshit

Academic Editor

PLOS One

Journal Requirements:

“This research was funded by U.S. National Science Foundation (grant number 0001024000) and Chinese Academy of Sciences Open Fund (grant number: 322967812).”

“The authors gratefully acknowledge the financial support from the U.S. National Science Foundation (grant number 0001024000) and the Chinese Academy of Sciences Open Fund (grant number 322967812).”

“This research was funded by U.S. National Science Foundation (grant number 0001024000) and Chinese Academy of Sciences Open Fund (grant number: 322967812).”

Reviewers' comments:

Reviewer's Responses to Questions

**Comments to the Author**

1. Is the manuscript technically sound, and do the data support the conclusions?

Reviewer #1: Yes

Reviewer #2: Partly

2. Has the statistical analysis been performed appropriately and rigorously?

Reviewer #1: Yes

Reviewer #2: Yes

3. Have the authors made all data underlying the findings in their manuscript fully available?

Reviewer #1: Yes

Reviewer #2: Yes

4. Is the manuscript presented in an intelligible fashion and written in standard English?

Reviewer #1: Yes

Reviewer #2: Yes

Reviewer #1: This paper demonstrates excellent performance in addressing multi-layer and multi-channel deposition defects. It has a rigorous logic, detailed professional simulations, and innovatively applies fuzzy control to enhance accuracy. However, several aspects require improvement

1、The English expression of the article contains grammatical errors and overly long and repetitive sentences, which may affect the readers' understanding.

2、The introduction and literature review section are too lengthy (with 25 references), but lack critical summaries.

3、The finite element simulation is overly simplified, and the heat source model neglects the precise values of deformation heat and latent heat of phase change, merely converting them into arc heat. This may lead to simulation errors. Additionally, information such as the sources of material properties (such as the thermal physical parameters of Q345 steel) needs to be provided.

4、Fuzzy logic has already been applied in welding control (refer to 24-25 for uncertain systems). The innovation of this paper lies in the adaptive extension of WAAM in specific scenarios, but it does not highlight the essential differences from existing strategies (such as adaptive slicing [17]). Please explain this.

Reviewer #2: Authors have focused on their work on ensuring the quality and dimensional accuracy of the components particularly in complex multilayer and multi-channel deposition processes. They demonstrate the control strategy to improve the forming accuracy and contributing to higher qualification rates for additively manufactured components. Reviewer has following points as illustrated below:

1. Novelty of this work is limited. Authors have to present the novelty in the introduction and abstract section.

2. Conclusion is very generic. Mention the numerical finding data in the conclusion section.

3. Authors can provide the comparison table parameters of the output and compare with existing findings.

4. Methodology is not accurately reported in the article.

5. Fuzzy logic part is not clearly mentioned. Illustrate at least key equations of the fuzzy logic in the article.

6. Add the recent 2025 literature and contrast the findings.

7. There are many typos error in the article.

8. Nearby equation 5 all sentences are wrong.

9. See line no 209, sentence is not correct

10. Manuscript is not clearly written. Many sentences are not readable.

11. Experimental aspects are missing in the manuscript.

**Do you want your identity to be public for this peer review?** For information about this choice, including consent withdrawal, please see our Privacy Policy

Reviewer #1: No

Reviewer #2: No

---

## [Author Response · Author response to Decision Letter 1]

7 Jan 2026

Dear Editor:

We sincerely thank the reviewers for their valuable time and insightful comments, which have greatly helped us improve the quality and clarity of our manuscript. We have carefully addressed all the concerns raised and have revised the manuscript accordingly.

Here below is our description on revision according to the reviewers comments

Part A( Comments from the Reviewers 1)

1. The reviewer’s comment: The English expression of the article contains grammatical errors and overly long and repetitive sentences, which may affect the readers' understanding.

The authors’s Answer: Thanks for your suggestion, the manuscript has undergone thorough language editing to correct grammatical errors, simplify complex sentences, and eliminate redundancies. The text has been refined to enhance clarity, readability, and academic tone.

2. The reviewer’s comment: The introduction and literature review section are too lengthy (with 25 references), but lack critical summaries.

The authors’s Answer: Thanks for your suggestion, the introduction section has been significantly streamlined. Several non-essential references have been removed, and the number of citations has been reduced to focus on the most relevant and recent works. A more critical synthesis of the literature has been provided, clearly identifying the research gap—specifically, the lack of real-time adaptive control strategies for MLMC deposition with variable channel counts—and positioning our work accordingly.

we have added the following descriptions:

"These methods can effectively enhance efficiency during the planning phase, but pre-set parameters are unable to resolve dynamic disturbances caused by heat accumulation and varying heat dissipation conditions during the WAAM manufacturing process. This often leads to unpredictable deviations between the actual formed component and the design model.

In current studies, feedback control strategies for layer geometry in WAAM remain insufficient. Most studies often focus on single-variable control, such as bead width or height[16-20]. For example, Doumanidis et al. [16] developed a multi-variable adaptive controller for the WAAM process, which is based on a generalized one-step-ahead control algorithm. Xiong et al. [17] proposed to use a single neuron self-learning controller to regulate the bead width during WAAM. Besides, Xia et al. [18] developed a Model Predictive Controller (MPC) to control head height during the WAAM process. However, MLMC deposition is a complex, multi-physics, strongly coupled, and parameter-time-varying process, making it difficult to establish an accurate analytical model. The number of deposition channels per layer in MLMC components varies (e.g., pyramid or hourglass structures), and traditional single-variable or fixed-structure controllers fail to adapt such variations." [revised manuscript page 2 line 43-55 ]

3. The reviewer’s comment: The finite element simulation is overly simplified, and the heat source model neglects the precise values of deformation heat and latent heat of phase change, merely converting them into arc heat. This may lead to simulation errors. Additionally, information such as the sources of material properties (such as the thermal physical parameters of Q345 steel) needs to be provided.

The authors’s Answer: We appreciate this technical feedback. In the revised manuscript (Section 2.1), we have added a clarification regarding the heat source modeling. It is now stated that, for practical numerical simulation, the latent heat of phase change and resistive heating are incorporated into the effective arc heat input due to the difficulty in their precise experimental quantification, and deformation heat is neglected due to its relatively minor contribution. We acknowledge this as a simplification and have included a note on its potential limitations. Furthermore, we have specified that the thermophysical properties of Q345 steel were taken from the standard material library in ANSYS, which is a common practice in such simulations. [revised manuscript page 3,4 line 113-133 ]

4. The reviewer’s comment: Fuzzy logic has already been applied in welding control (refer to 24-25 for uncertain systems). The innovation of this paper lies in the adaptive extension of WAAM in specific scenarios, but it does not highlight the essential differences from existing strategies (such as adaptive slicing [17]). Please explain this..

The authors’s Answer: Thank you for raising this point. We have now explicitly highlighted the core innovation of our work in both the Abstract and Introduction. While fuzzy logic is indeed applied in control systems, our contribution is the development of an inter-layer control strategy that can adapt in real-time to a variable number of deposition channels (e.g., in pyramidal or funnel structures). This distinguishes it from:

a) Offline slicing optimization methods (e.g., Ref. [17], now Ref. [15] in the revised list), which lack real-time correction capability.

b) Traditional fuzzy controllers designed for fixed-geometry or single-variable control in welding.

This distinction is further elaborated in the Discussion section, where we compare our real-time adaptive approach with offline optimization and material-modification strategies:

"Unlike material modification approaches that enhance intrinsic properties through alloying elements[12], the proposed strategy actively compensates for deformation caused by uneven temperature field and residual stress by adaptively regulating the stacking speed. Furthermore, previous research focused on computational efficiency through slicing algorithm optimization [13-15], this study emphasizes real-time adaptive control during the deposition process. In practical applications, the method provides a valuable technical support for controlling the precision of complex components in rapid manufacturing and remanufacturing. It can be directly integrated into existing intelligent additive manufacturing systems to improve process stability and product consistency.

While the current study demonstrates promising results, some limitations should be acknowledged. The control strategy was validated under specific deposition conditions and material (Q345 steel). Its performance with other materials (e.g., aluminum alloys or high-strength steels) or under significantly different thermal regimes deserves further investigation. Additionally, the controller's parameters were tuned for the experimental setup described; generalizing these parameters for broader industrial applications may require additional adaptive mechanisms".

Part B( Reviewer 2)

1. The reviewer’s comment: Novelty of this work is limited. Authors have to present the novelty in the introduction and abstract section.

We have revised both the Abstract and Introduction to clearly articulate the novelty. The Abstract now states: "a novel real-time inter-layer control strategy based on fuzzy logic was proposed. A fuzzy logic controller was designed to adaptively regulate the stacking speed of each deposition channel by utilizing the real-time height deviation and its rate of change as inputs." for MLMC deposition.

A more critical synthesis of the literature has been provided, clearly identifying the research gap—specifically, the lack of real-time adaptive control strategies for MLMC deposition with variable channel counts—and positioning our work accordingly. we have added the following descriptions:

"These methods can effectively enhance efficiency during the planning phase, but pre-set parameters are unable to resolve dynamic disturbances caused by heat accumulation and varying heat dissipation conditions during the WAAM manufacturing process. This often leads to unpredictable deviations between the actual formed component and the design model.

In current studies, feedback control strategies for layer geometry in WAAM remain insufficient. Most studies often focus on single-variable control, such as bead width or height[16-20]. For example, Doumanidis et al. [16] developed a multi-variable adaptive controller for the WAAM process, which is based on a generalized one-step-ahead control algorithm. Xiong et al. [17] proposed to use a single neuron self-learning controller to regulate the bead width during WAAM. Besides, Xia et al. [18] developed a Model Predictive Controller (MPC) to control head height during the WAAM process. However, MLMC deposition is a complex, multi-physics, strongly coupled, and parameter-time-varying process, making it difficult to establish an accurate analytical model. The number of deposition channels per layer in MLMC components varies (e.g., pyramid or hourglass structures), and traditional single-variable or fixed-structure controllers fail to adapt such variations." [revised manuscript page 2 line 43-55 ]

The Introduction concludes by stating that the paper aims to "propose and validate an inter-layer control strategy based on fuzzy logic that adapts to variations in the number of deposition channels.".

2. The reviewer’s comment: Conclusion is very generic. Mention the numerical finding data in the conclusion section.

The authors’s Answer: Thanks for your suggestion. The Conclusion section has been substantially revised to include specific key findings:

"Finite element simulation of a representative pipeline joint revealed crucial thermal behavior during MLMC deposition. The results quantified significant residual stress concentrations at structural junctions (up to 168 MPa) and identified subsidenc (-3.75×10⁻⁵ mm) and inclination tendencies in branch pipes. To adaptively compensate for the identified thermal deformation, a fuzzy logic controller (FLC) was designed to dynamically adjust the deposition speed of each channel based on real-time layer height deviation and its rate of change. Meanwhile, the controller enable to adapt structures with variable channel counts (pyramidal and funnel geometries) without requiring precise analytical models of highly nonlinear processes. Experimental validation confirmed the effectiveness of the proposed controller under various structural configurations. For a simple 20-layer vertical structure, the maximum height deviation was constrained to 0.13 mm. For 60° and 70° inclined structures, the controller ensured stable layer growth, with overall deviations fluctuating near zero. Furthermore, for a complex funnel structure with dynamically varying channel numbers, the strategy maintained the top-layer error within 0.25 mm without significant edge subsidence or inclination."[revised manuscript page12, line 343-353]

3. The reviewer’s comment: Authors can provide the comparison table parameters of the output and compare with existing findings.

The authors’s Answer: Thanks for your kind suggestions, which is valuable for improving the accuracy of the manuscript. we integrated a detailed comparative analysis into the Discussion section (Section 4). We compare our strategy's focus on real-time adaptive process control with previous works focused on offline slicing optimization [14, 15] and material modification [12], clearly stating the advantages of our approach in handling dynamic deposition conditions.

"Unlike material modification approaches that enhance intrinsic properties through alloying elements[12], the proposed strategy actively compensates for deformation caused by uneven temperature field and residual stress by adaptively regulating the stacking speed. Furthermore, previous research focused on computational efficiency through slicing algorithm optimization [14], this study emphasizes real-time adaptive control during the deposition process. In practical applications, the method provides a valuable technical support for controlling the precision of complex components in rapid manufacturing and remanufacturing. It can be directly integrated into existing intelligent additive manufacturing systems to improve process stability and product consistency.

While the current study demonstrates promising results, some limitations should be acknowledged. The control strategy was validated under specific deposition conditions and material (Q345 steel). Its performance with other materials (e.g., aluminum alloys or high-strength steels) or under significantly different thermal regimes deserves further investigation. Additionally, the controller's parameters were tuned for the experimental setup described; generalizing these parameters for broader industrial applications may require additional adaptive mechanisms."[revised manuscript page12, line 326-337]

4. The reviewer’s comment: Methodology is not accurately reported in the article.

The authors’s Answer: Thanks for your constructive suggestion, which is highly appreciated. We have comprehensively rewritten Section 2 (Materials and Methods) to ensure accurate and complete reporting. This includes clearer descriptions of the FEA setup (boundary conditions, meshing strategy, solver settings) and a step-by-step explanation of the fuzzy controller design and implementation logic. [revised manuscript page 3-6 ].

5. The reviewer’s comment: Fuzzy logic part is not clearly mentioned. Illustrate at least key equations of the fuzzy logic in the article

The authors’s Answer: Thanks for your kind suggestion, We have made changes in the revised manuscript. Section 2.2 has been expanded for clarity. We now explicitly define:

(1) Inputs: Height deviation and its rate of change (Δe).

(2) Output: The change in speed increment ( ).

(3) The fuzzy rule table (Table 1) is presented with a clearer explanation.

(4) The core equations for calculating , (Eq. 5), and the final stacking speed (Eq. 6) are provided and explained in the context of the control loop.

[revised manuscript page 5 ].

6. The reviewer’s comment: Add the recent 2025 literature and contrast the findings.

The authors’s Answer: Thanks for your suggestion. which is valuable for improving the accuracy of the manuscript. We have added relevant 2025 references to the updated reference list and have discussed recent trends in the Introduction to better situate our work within the current research.

7. The reviewer’s comment: There are many typos error in the article.

The authors’s Answer: We apologize for the trouble caused by our mistake to the reviews. The manuscript has been meticulously proofread by the authors and checked with professional editing tools to correct all typographical and grammatical errors.

8. The reviewer’s comment: Nearby equation 5 all sentences are wrong.

The authors’s Answer: The text surrounding Equation 5 has been completely rewritten to ensure logical flow and technical accuracy in describing the calculation of height deviation and its change.

"Assuming the deposited layer inside a certain part is composed of n deposited channels, the preset forming height of the n-th welding channel in the m-layer is after stacking, and the actual forming height is . The actual formed height deviation and the change in height deviation at that position are given by Equation (5):

(5)

Directly applying control based on may cause instability due to the inconsistent stacking speed of different layers in the deposited layer. Therefore, this study uses the change amount of velocity increment for control, which is accumulated layer by layer to determine the optimal velocity offset. By inputting and into the fuzzy logic inference machine, the change amount of the accumulation velocity increment of the next layer at that position of the weld deposit is obtained. This incremental change is added to the velocity increment calculated for the current deposition channel to yield the velocity increment for the next layer. Finally, the stacking speed of the deposition channel in the next layer is modified, as expressed in Equation (6)."[revised manuscript page 5 ].

9. The reviewer’s comment: See line no 209, sentence is not correct.

The authors’s Answer: We apologize for the trouble caused by our mistake to the reviews. Based on the revised manuscript's line numbering, the corresponding sentence has been identified and corrected for clarity and correctness.

"By inputting and into the fuzzy logic inference machine, the change amount of the ac

---

## [Decision Letter · Decision Letter 1]

1 Feb 2026

Multi-layer and Multi-channel Deposition Defects and Inter-layer Control Strategies in Additive Manufacturing Materials

PONE-D-25-58463R1

Dear Dr. Li,

We’re pleased to inform you that your manuscript has been judged scientifically suitable for publication and will be formally accepted for publication once it meets all outstanding technical requirements.

Kind regards,

Dr. Mithilesh K. Dikshit

Academic Editor

PLOS One

Additional Editor Comments (optional):

Reviewers' comments:

Reviewer's Responses to Questions

**Comments to the Author**

Reviewer #1: All comments have been addressed

Reviewer #2: All comments have been addressed

2. Is the manuscript technically sound, and do the data support the conclusions?

Reviewer #1: Yes

Reviewer #2: Yes

3. Has the statistical analysis been performed appropriately and rigorously?

Reviewer #1: Yes

Reviewer #2: N/A

4. Have the authors made all data underlying the findings in their manuscript fully available?

Reviewer #1: Yes

Reviewer #2: Yes

5. Is the manuscript presented in an intelligible fashion and written in standard English?

Reviewer #1: Yes

Reviewer #2: Yes

Reviewer #1: (No Response)

Reviewer #2: All comments are addressed. The revised paper can be accepted for the publication in the PLOS ONE Journal.

**Do you want your identity to be public for this peer review?** For information about this choice, including consent withdrawal, please see our Privacy Policy

Reviewer #1: No

Reviewer #2: **Yes:** nitish kumar

---

## [Editor Report · Acceptance letter]

PONE-D-25-58463R1

PLOS One

Dear Dr. Li,

I'm pleased to inform you that your manuscript has been deemed suitable for publication in PLOS One. Congratulations! Your manuscript is now being handed over to our production team.

Kind regards,

on behalf of

Dr. Mithilesh K. Dikshit

Academic Editor

PLOS One